# Normalisation process theory and the implementation of a new glaucoma clinical pathway in hospital eye services: Perspectives of doctors, nurses and optometrists

**Simon Read**[1]*, **James Morgan**[2], **David Gillespie**[3], **Claire Nollett**[3], **Marjorie Weiss**[4], **Davina Allen**[1], **Pippa Anderson**[5], **Heather Waterman**[1]

1 School of Healthcare Sciences, Cardiff University, Cardiff, Wales, 2 Cardiff and Vale University Health Board, Cardiff, Wales, 3 Centre for Trials Research, Cardiff University, Cardiff, Wales, 4 School of Pharmaceutical Sciences, Cardiff University, Cardiff, Wales, 5 Swansea Centre for Health Economics, Swansea University, Swansea, Wales

* readsm@cardiff.ac.uk

## Abstract

### Background

Normalisation process theory reports the importance of contextual integration in successfully embedding novel interventions, with recent propositions detailing the role that 'plasticity' of intervention components and 'elasticity' of an intended setting contribute. We report on the introduction of a clinical pathway assessing patient non-responsiveness to treatment for glaucoma and ocular hypertension. The aim of this study was to assess the feasibility of implementing the Cardiff Model of Glaucoma Care into hospital eye services, identifying any issues of acceptability for staff through the filter of normalisation process theory.

### Methods

A prospective observational study was undertaken in four hospital eye services. This incorporated detailed qualitative semi-structured interviews with staff (n = 8) to gather their perceptions on the intervention's usefulness and practicality. In addition, observational field notes of patient and staff consultations (n = 88) were collected, as well as broader organisational observations from within the research sites (n = 52). Data collection and analysis was informed by the normalisation process theory framework.

### Results

Staff reported the pathway led to beneficial knowledge on managing patient treatment, but the model was sometimes perceived as overly prescriptive. This perception varied significantly based on the composition of clinics in relation to staff experience, staff availability and pre-existing clinical structures. The most commonly recounted barrier came in contextually integrating into sites where wider administrative systems were inflexible to intervention components.

**Data Availability Statement:** This is a qualitative study confined to a limited locality (Wales, UK). Our ethics approval was granted based on the

anonymity of the individuals consenting to participate. Participants were drawn from a small group of health care professionals in specific roles, particularly those delivering the intervention. Providing additional information beyond the carefully selected anonymised quotations that support the findings in the manuscript, such as additional excerpts from anonymised/ pseudonymised transcripts, full transcripts or the full data set, would enable others to identify the participants by the way they describe their interactions with the intervention and associated activities. Furthermore our ethics approvals were based upon statements in the participant information sheets and consent forms that specifically referred to anonymised quotations from transcripts being used. As such the participants did not consent to information beyond those quotations being made publically available. Ethics approval was granted by West Midlands – Black Country NHS Research Ethics Committee on 16 November 2017, IRAS Project ID: 232242. Information on the data underpinning the results presented here, including how to access them, can be found in the Cardiff University data catalogue at: http://doi.org/10.17035/d.2021.0135330917 - opendata@cardiff.ac.uk.

**Funding:** JM was awarded the funding for this research alongside several co-applicants forming some of the research team (HW, DG, CN, MW, DA, PA). This research was funded in full by the Health and Care Research Wales Research for Patient and Public Benefit fund, ref: RfPPB-16a-1296. The funders had no role in study design, data collection and analysis, decision to publish, or preparation of the manuscript.

**Competing interests:** The authors have declared that no competing interests exist.

## Conclusions

Flexibility will be the key determinant of whether the clinical pathway can progress to wider implementation. Addressing the complexity and variation associated with practice between clinics required a remodelling of the pathway to maintain its central benefits but enhance its plasticity. Our study therefore helps to confirm propositions developed in relation to normalisation process theory, contextual integration, intervention plasticity, and setting elasticity. This enables the transferability of findings to healthcare settings other than ophthalmology, where any novel intervention is implemented.

## Introduction

The importance of context to the implementation of complex interventions has come into increasing focus in recent years [1]. Within the healthcare setting, numerous studies have highlighted the role that context plays in whether an implementation is successful, generally noting such features as leadership, organisational culture, communication, resources, monitoring and feedback and implementation champions [2–4]. At a more conceptual level, normalisation process theory (NPT) also states the importance of *contextual integration* when understanding how and why an intervention becomes embedded within nursing or healthcare settings [5–8]. Here, rather than an emphasis on contextual features, recent thinking has highlighted the role that the 'elasticity' of an intervention's setting and the 'plasticity' of its components influences over implementation [5]. Within this the proposition is that the more elastic an intervention setting, and the more plastic the components of an intervention, the less strain there will be on those implementing it and therefore an increased likelihood it will be implemented [5]. The practical implications of this for implementing healthcare interventions are wide-ranging, including aspects of intervention design and planning, as well as the value of understanding variation within and between specific contexts.

## Background

We report on a study investigating non-responsiveness to treatment for the degenerative eye condition, glaucoma. In the United Kingdom, glaucoma affects around 600,000 people, although it is suggested that a further 300,000 cases may remain undiagnosed and future projections show rises in line with the wider ageing population [9]. The most common treatment for the condition is the instillation of daily eye drops that reduce the build-up of intraocular pressure (IOP) responsible for damaging the optic nerve [10]. However, previous research has demonstrated that not all patients respond physiologically to treatment and that this can remain undiagnosed for several months placing patients at risk of visual field loss [11].

Our study assessed whether a new evidence-based clinical pathway intervention, the Cardiff Model of Glaucoma Care (CMGC, Fig 1) [12], was feasible and acceptable to doctors, nurses and optometrists in four hospital eye services. The purpose of the CMGC intervention was to establish whether patients were responsive to their treatment when first initiated. This was achieved by staff instilling an eye drop at the clinic and IOP measurement four hours' later. Four weeks later patients returned to have their IOP re-measured to determine whether they were still responding to treatment.

Evaluation of the implementation saw overlaps with NPT, commonly applied to complex interventions [6–8]. While the CMGC was implemented successfully across all sites for the

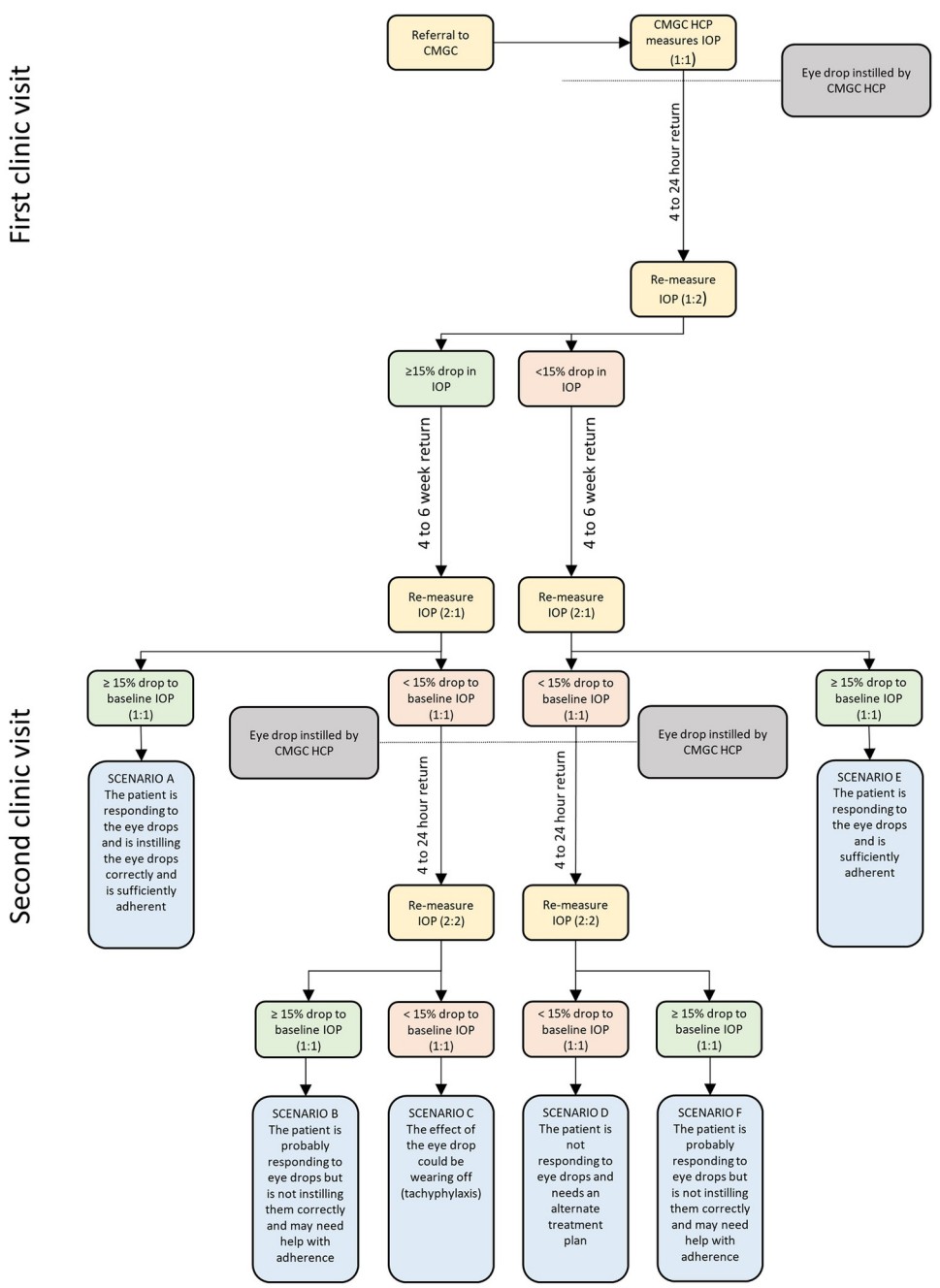

**Fig 1. Final Cardiff Model of Glaucoma Care Algorithm [12].**

prescribed research period, the level of strain perceived by those enacting it varied from site to site. Predominantly, the themes of NPT were more tangible where the implementation was felt to be more onerous by those enacting it, with the smoother implementations offering guidance on how certain contextual challenges for one site may be less problematic elsewhere. Developed over time, the four key interrelated constructs within NPT, outlined in Table 1, offer a means of understanding factors that enable or disable successful implementations. For instance, numerous studies cite specific issues associated with *coherence* [13–15] while others

**Table 1. Normalisation process theory constructs & description [19].**

| Normalisation Process Theory Constructs | Description |
|---|---|
| *Coherence* | Refers to the 'sense-making work' carried out by individuals and collectively within organisations when implementing a new set of practices, understanding the purpose and benefits of them being a pre-requisite to their success. **Subcomponents** Differentiation; communal specification; individual specification; internalization. |
| *Cognitive Participation* | Refers to the 'relational work' carried out to 'build and sustain a community of practice' around an intervention, including the involvement of key stakeholders to drive it forwards. **Subcomponents** Initiation; enrolment; legitimation; activation. |
| *Collective Action* | Refers to the 'operational work' required to enact a new set of practices, such as staff resourcing, equipment availability and other issues specific to local contexts. **Subcomponents** Interactional workability; relational integration; skill set workability; **contextual integration** (*including plasticity and elasticity*). |
| *Reflexive Monitoring* | Refers to the 'appraisal work' in understanding how a new set of practices affects those engaging with them. **Subcomponents** Systematisation; communal appraisal; individual appraisal; reconfiguration. |

focus more on issues of *cognitive participation* [16–18]. While encompassing other NPT constructs and subcomponents, this article focuses on the implementation of the CMGC predominantly through the subcomponent of *contextual integration* and the related, relatively underexplored notions of plasticity and elasticity. Additionally, discussions will be opened on the extent that plasticity can be built into interventions before compromising their clinical quality, and how an intervention setting's perceived elasticity can vary significantly between locality and job role.

## Materials and methods

### Aims

The research objectives for the qualitative aspects of the study were:

1. Is the CMGC acceptable to healthcare staff?

2. How might the CMGC become embedded into practice?

3. In what ways does NPT help to understand the barriers and facilitators to implementing the CMGC?

Together these study aims looked to identify issues associated with how the CMGC was implemented, primarily from the perspectives of the healthcare staff working with it. The application of NPT also afforded opportunity to assess which elements of the intervention, if any, were deemed particularly problematic for staff and how these may be best resolved.

### Design

The overarching study was a single arm non-randomised prospective observational study investigating the CMGC intervention outlined in Fig 1 [12]. For the qualitative elements, an ethnographic approach [20] incorporating observations and interviews was adopted exploring

how healthcare staff responded to the introduction of the CMGC pathway and how the context of the implementation affected their responses.

## The CMGC intervention

Key features of the CMGC intervention were additional patient appointments built in to identify those with less than 15% reduction in IOP, deemed as non-responsiveness to treatment, when compared to their baseline IOP measurement [12]. The staff followed a prescribed algorithm which detailed how to proceed at each of the two clinic visits over four weeks [12]. After being provided information on the study and the new care pathway, consented patients arrived into clinic for baseline IOP measurement before having an IOP-reducing eye drop instilled on their behalf by a member of staff. Four hours' later, the patient arrived back for another appointment where their IOP was re-measured and assessed for any noticeable effect. Patients were informed of their results on each occasion. Another set of appointments was then arranged four weeks' later where this process would be repeated; a further re-measurement of IOP, followed by the instillation of an eye drop by a staff member. Again, four hours' later, the effect of this drop on the IOP was assessed, establishing whether patients responded to eye drops and/or needed support with adherence before making a decision on the ongoing treatment plan.

## Sample/Participants

After being provided information on the study, participants were purposively enrolled into the CMGC from four routine glaucoma clinics (see Table 2 for pseudonymised site characteristics).

Staff were purposively recruited to assist with implementing the CMGC intervention, carrying out its procedures and offering their views on any barriers or facilitators to a wider implementation. Inclusion criteria for these participants were: being employed by one of the research sites; and being either a doctor, nurse, orthoptist or optometrist. Staff invited to work within the intervention were selected based on their involvement with existing services and in conjunction with the participating health boards.

**Table 2. Eye clinic site characteristics.**

| Clinic | Description |
|---|---|
| Birch Clinic | • 4 CMGC staff: general ophthalmic nurses, consultant ophthalmologist and specialist doctor.<br>• Secondary care consultant-led outpatient clinic with junior doctors assisting.<br>• Clinic held across one full day with same staffing AM to PM. |
| Oak Clinic | • 2 CMGC staff: consultant ophthalmologist and specialist optometrist.<br>• Secondary care consultant-led outpatient clinic with optometrists assisting.<br>• Clinic held for half day in AM. |
| Cedar Clinic | • 4 CMGC staff: consultant ophthalmologist, glaucoma ophthalmic nurses.<br>• Secondary care nurse-led clinic with consultant remote review.<br>• Clinics held across one full day with same staffing AM to PM. |
| Maple Clinic | • 5 CMGC staff: consultant ophthalmologist, doctors and orthoptists.<br>• Secondary care consultant-led clinic with orthoptists assisting.<br>• Clinic held for half day in PM. |

## Data collection

This article reports two different sources of qualitative data. In assessing the extent to which the CMGC pathway was acceptable to participants, we carried out observations, as well as semi-structured interviews with the healthcare staff responsible for implementing the intervention. All data collection took place between June 2018 and March 2019.

**i) Clinical observations.** Researchers independent from the clinical team sat in the CMGC consultations between staff and patients, documenting any procedural, professional or behavioural issues associated with the appointments, as well as any variations in practice. In total, 88 consultations were observed, each lasting between 10 and 20 minutes, incorporating 50 patients and 10 staff. These observations were documented on a study template consisting of headings for the site, date of observation, staff discipline and patient reference numbers, length of observation, and free text sections for summary and narrative description. Alongside consultations, wider observational data were collected week-by-week around issues associated with the implementation not evident within the appointments themselves. These included organisational or technological issues that had potential to influence whether the intervention could be implemented more widely. A total of 52 of these observational field notes were taken over the duration of the study covering the majority of research visits.

**ii) Semi-structured interviews.** Interviews were sought with staff involved with the CMGC intervention, recruiting eight participants from the wider sample of 16 contributors. Purposive sampling was carried out to interview those who worked most closely with the intervention in each site, given some participants were recruited as cover for primary staff and interacted with the CMGC much less. A schedule was developed integrating core constructs of NPT, with the NoMAD instrument acting as a broad topic guide [21], as well as considering evidence gathered from local observations. The schedule was intentionally loose, being careful not to force responses into the NPT framework through open-ended questions. These focussed on whether the CMGC met interviewees' expectations in its clinical effect and what they perceived to be the barriers and facilitators to future implementations. Interviews were all audio-recorded one-on-one, lasting between 20 and 80 minutes.

## Ethical considerations

The study received ethical approval from West Midlands–Black Country NHS Research Ethics Committee on 16 November 2017, IRAS Project ID: 232242. All participants were given information sheets about the study prior to gaining their written informed consent and all practices followed the guidelines of the Declaration of Helsinki [22].

## Data analysis

The analytical process followed the principles of framework analysis, commencing with a period of familiarisation where transcripts, field notes and recordings were revisited by the team to identify consistently emerging themes [23]. Both sets of qualitative data were analysed together to allow cross-comparison between them. With these preliminary themes established, the next phase of analysis involved the development of a framework to make collective sense of them. Many of the study's a priori concerns relating to the normalisation of the intervention factored into this process through the core constructs and subcomponents of NPT, although analysis was also sensitive to localised themes emerging from the data.

### Rigour

Data trustworthiness was enhanced by the collection and analysis of multiple data sources: longitudinal organisational observations; clinical consultation observations; and staff and patient interviews [24]. Furthermore, the input of the experienced research team during coding to the developed framework informed the indexing of datasets to applicable themes. This was done with the assistance of QSR International NVivo software to organise data, with regular check-ins between researchers and senior investigators to ensure agreement on where data fitted within the developing themes [25]. Once coding was complete, the final thematic framework of data was mapped and interpreted collectively by the research team, establishing the key themes relating to the implementation and how these interrelated [23].

## Results

The recruited sample resulted in a range of staff being observed and interviewed: nurses, junior doctors, orthoptists, optometrists and consultants. Findings are presented in themes according to key variations in the clinical context that most affected acceptability of the CMGC and its wider implementation. These were identified as staff availability, CMGC staff specialism, and pre-existing clinical structures.

Data analysis established the salience of several NPT core constructs, as well as their sub-components, in explaining variabilities in implementation. The overarching NPT construct most relevant to our findings was *collective action*, alongside its sub-component of *contextual integration*. Nevertheless, as has been summarised elsewhere, the various constructs and sub-components have potential to overlap and, as such, findings in these other areas are also reported [18]. For instance, the theme of staffing for the CMGC highlights the relevance of the construct of *cognitive participation*, specifically issues associated with *initiation* and *enrolment*. Job specialism, however, aligns with the constructs of *coherence* and *collective action* and sub-components of *legitimation* and *skill set workability*. Finally, pre-existing clinical structures develops the emerging debate on plasticity and elasticity for healthcare interventions associated with the NPT subcomponent of *contextual integration*.

### Staffing for CMGC

The research sites each had varying staff numbers, job role, clinic timings and follow-up strategy (Table 2). These had potential to indirectly influence key aspects of how the CMGC was perceived by staff implementing it. Each research site was given autonomy when selecting staff to be involved with the implementation of CMGC, although each clinic required a consultant ophthalmologist to act as Principal Investigator. Based on each site's pre-existing staff composition the number of staff allocated to the CMGC varied. In three of the clinics, the number of CMGC staff within each site was not viewed problematically in clinical observations, with capacity rarely provoking delays to appointments. However, the limited number of staff enrolled in the CMGC from Oak Clinic, alongside broader departmental capacity issues led to CMGC recruitment being halted and ultimately abandoned as key staff resigned or had long periods of leave:

> "*As OH302 (the assigned CMGC clinician) leaves, there is little clarity on how the CMGC will be hosted in Oak Clinic after that point*"

> **Week-by-Week Observation 180716; Oak Clinic**

Efforts to identify alternative CMGC staff to facilitate the intervention for this site were unsuccessful. This linked to the NPT construct of *cognitive participation*, and particularly the subcomponents *initiation* and *enrolment*. Both are concerned with the correct staff being in place to build and maintain the set of new practices around the intervention. Within the context of Oak Clinic, this was evidently not the case; at the point of initiation greater staff cover would have ideally been enrolled to mitigate the issues of any departures once the intervention was active.

## CMGC staff and job specialism

Each of the CMGC clinics offered a different dynamic of staff. For most clinics, the CMGC was staffed by those with specific knowledge of existing glaucoma care pathways. However, Birch Clinic provided a blend of glaucoma specialists working alongside general ophthalmic nurses, with the latter predominantly providing patient-facing care.

In terms of the CMGC procedures, these staff were all comfortable performing procedures such as IOP measurement and eye drop installation. That said, based on a less extensive understanding of glaucoma, certain facets of consultations left them feeling illegitimate to perform their roles: *"If the patient asked me anything technical, I would have struggled"* **BH303**; *"Glaucoma is not a part of my domain"* **BH302**. These staff felt that they had insufficient glaucoma-specialist knowledge and should it be required from them, they would feel inadequate to give it. This theme aligns closely with the NPT subcomponents of *legitimation* (*coherence*) and *skill set workability* (*collective action*). The clinicians, while legitimately placed in some respects based on their broader ophthalmic knowledge and technical capabilities, felt that in practice discussing the condition in depth with patients was not consistent with their natural skill set.

When this is compared to Cedar Clinic or Maple Clinic where the patient-facing CMGC staff were glaucoma-specific ophthalmic nurses or orthoptists, these themes appeared less of an issue. Staff who had greater tacit knowledge of the treated condition, e.g. consultants, glaucoma nurses, as well as the practical skills to perform key clinical procedures, reported more comfort with the CMGC: *"That's part of my standard care. . .that wasn't any different than I do normally"* **CH302**; *"The clinical stuff, the clinical side of actually doing it is fine"* **MH306**.

Beyond this, those staff with more formal knowledge of glaucoma were also observed to have a more instinctive understanding of the benefits of the CMGC, offering perspectives on the future use of the intervention as well as the usefulness of the data: *"If I knew on day one who was going to work on drops and who wasn't that would be so beneficial"* **OH301**; *"If the patient responds to the dose given. . .and that dose is confirmed as being effective on intraocular pressure at the four hour mark. . .that is a premise on which you can build the whole of the rest of your care"* **CH301**. Tying to the NPT construct of *coherence*, those staff with less tacit knowledge of the glaucoma care pathways and the condition itself, appeared less able to perform sense-making work. This extended to differentiating between the CMGC and standard care, as well as ultimately internalising the intervention within their day-to-day duties: *"I mean this was for a given period of time, so I was quite happy to see those patients ad hoc"* **BH302**. By the end of the study period, those staff with less glaucoma expertise did not feel their involvement with the intervention would be maintained. As such, while performing the duties within the confines of the research was acceptable, issues of *internalisation*, *skill set workability* and *legitimation* all had potential to negatively influence long-term staff response to the CMGC.

## Pre-existing clinical structures

One of the more commonly reported issues was a perceived rigidity in the CMGC's arrangement of appointments. This included prescribed four-hour follow-up appointments for IOP

re-measurement following the instillation of an eye drop, as well as four-week follow-up appointments to monitor patient efficacy and adherence with instilling drops themselves. For each clinic, these requirements prompted varying levels of operational reorganisation so that the CMGC pathway could be integrated. By example, Maple Clinic hosted initial CMGC appointments on an ad hoc basis during administrative sessions on a weekday morning when clinicians were free. However, four hours' later these same clinicians were then hosting an afternoon clinic with limited capacity for any CMGC appointments or IOP re-measurement: *"The trouble came when we were, after the four hours. . .trying to identify somebody to do the measurements"* **MH306**. Likewise, Oak Clinic encountered similar issues with the two designated CMGC clinicians available during a morning clinic but either off site or in surgical theatre during the afternoon session:

> "*Patient arrived at 14:25 for return appointment at 14:30. OH301 was in theatre with a trauma patient. . .OH301 arrived at 14:55 based on complications in theatre, apologising to patient for the delay*"

**Clinical Consultation 181001; Oak Clinic**

While Oak Clinic and Maple Clinic were able to host CMGC appointments, this often required staff fitting them in alongside other clinical commitments, or delays to appointments based on lack of staff availability. However, within Birch Clinic and Cedar Clinic, where the same staff were available from morning to the afternoon, this problem was much less noticeable. This highlights the importance of NPT subcomponent *contextual integration* for the CMGC, with structural variations in timetabling and staffing providing examples where the CMGC operated relatively smoothly alongside those where it was significantly more problematic.

Recent debates over *contextual integration* within NPT, and the plasticity and elasticity of an intervention and setting, suggest that flexibility within both ease an implementation's progress [5]. Regarding the CMGC, some participants reported that both the prescribed pathway, as well as restrictions within their respective health boards, made future implementations difficult. The clinics encountering issues with four-hour follow-up appointments also reported issues relating to the prescribed four-week follow-up appointments:

> "*Most places are working at least six weeks ahead. So I know that my clinics for the next six weeks are already booked. So therefore, if we identify somebody today who needs to go onto treatment, when do I bring them back*?"

**MH306**

Administrative procedures in each site were reported to restrict the ease of embedding the CMGC within standard care in the long term. This quoted clinician from Maple Clinic highlighted an inelasticity within the appointments system, based on existing processes guiding appointments towards six rather than four-week follow-ups. Other sites also reiterated this point: *"That's the same across Wales, you'll find that most patients that you want to see will appear back at six weekly intervals, but if you put in for seven weeks you won't see them for months"* **OH301**. Limitations within an intervention's intended setting would ideally have been identified during earlier stages of design and planning, suggesting that stakeholders with more detailed oversight of such administrative issues may have been overlooked during this process. This inclusive co-production of an intervention's components with staff localised in each setting links back to the NPT subcomponents of *enrolment* and *initiation*. While every

effort was made to identify organisational and technical issues in advance of implementation, the issues of follow-up periods was not highlighted at this stage meaning there was a lost opportunity for greater plasticity within the CMGC.

Beyond this, though, these findings in relation to pre-existing clinical contexts, and their perceived elasticity, are pivotally important to how an intervention may be perceived by staff. When staff felt empowered to enact organisational changes, it was noticeable that their local setting was perceived to have much greater elasticity: *"It's a capacity variance requirement rather than a huge amount of additional activity"* **CH301**; *"Yes, the system's a bit clunky. . .it's getting better as time goes on, is what I think I would say, it's not as bad as it was"* **BH301**. These were both more senior members of staff for whom the systems and setting around them did not appear to be significant barriers. This was a notable contrast from those staff who perceived the setting and, consequently the CMGC itself, as being inflexible: *"I think it is far too interventionist with a complete blanket of people too soon"* **MH307**. In terms of NPT, this has implications for how plasticity and elasticity are understood in relation to contextually integrating interventions into healthcare settings.

## Discussion

A core aim of this study was to understand the practicalities and acceptability of implementing the CMGC through the lens of NPT. The findings highlighted that NPT constructs and sub-components in the implementation of a complex healthcare intervention were applicable to staff and provide a deeper understanding of the NPT sub-components of setting elasticity and intervention plasticity than previously articulated [5]. However, like previous studies, we identified that large elements of NPT were more transparently applied to those staff implementing an intervention rather than to patients [18]. It is worth noting from our previous research, that the retention rate of patients for all CMGC study procedures ran at 94.3% [12]. Likewise, the majority of those approached to partake in the study also decided to do so, suggesting that the extra burden of two further clinic appointments was largely acceptable 12. From the present study, for staff there appeared a great deal more nuance in how the CMGC was perceived, with NPT offering a framework to understand how certain procedures were carried out relatively easily, while other aspects were felt to jeopardise the implementation.

Staff generally perceived the clinical information gathered from the CMGC positively and were also largely comfortable with all the required clinical procedures. On this basis, the NPT constructs of *coherence* and *cognitive participation* were largely felt to be well negotiated, enabling most procedures. The division of glaucoma and non-glaucoma specialists in Birch Clinic highlighted the importance of these themes in planning for the CMGC. While experienced ophthalmic nurses, those involved with the intervention day-to-day were uncomfortable discussing specifics of the condition with patients. Issues relating to these themes of *legitimation* and *skill set workability* are relatively common in studies of implementations, often in relation to specific tasks or activities felt inappropriate to staff [26,27]. Where such issues are encountered, there has generally been calls for further training to mitigate them. However, this would only be partially effective in this instance, given that the staff were never truly part of the glaucoma service within the health board. Instead, identifying alternative staff with a broader tacit knowledge of the treated condition would benefit future iterations.

Perhaps the more pertinent issues regarding implementation in this instance emerged in relation to the NPT subcomponent *contextual integration*. This centred on pre-existing systems of standard care that were perceived to be beyond staff control, such as the booking of appointments, lack of resource and the busyness of existing clinics. In each instance, the intervention was commonly regarded as provoking more problems through the rigid requirement

for additional appointments, the scheduling of these appointments into existing, non-compatible booking systems, and the reorganisation of clinical staff. The issue of context, its dynamic properties and conceptual terrain, has been debated extensively in the field, with the relationship between an intervention and its local setting shown to be a key factor in determining its success and sustainability [26,27]. In order to be integrated successfully into local contexts, May et al [5] highlight how an intervention's components may require normative and/or relational restructuring that enables greater user discretion. Within this, the authors highlight the importance of flexibility within an intervention's elements that enable its users to shape them within their own specific contexts:

> "*When intervention components are inflexible and rigidly applied, they require high levels of commitment from their users. . .these reduce the room for manoeuvre available to participants in implementation processes and mean that the transportability of intervention components between settings is inhibited*"

[5]

For the CMGC intervention, it was clear that certain components were problematic for staff to integrate into their specific care settings, illustrating the impact of an intervention without the *plasticity* to be easily accommodated. Largely, this was associated with the four-hour, same-day appointments, but they were also noticeable in the four-week follow-ups. While successfully implementing the CMGC in each of the sites [12], during interview some staff from certain clinics expressed frustration that the clinical benefits of the intervention were obfuscated by its prescriptiveness. In some instances, this was attributed to the timing of appointments running contrary to pre-existing systems, whereas on other occasions the organisational systems themselves were perceived to be inelastic and problematic. This demonstrates the counter importance of *elasticity* of the setting in which the intervention is implemented.

Numerous studies have identified similar, context-specific issues with implementation. A recent review outlined their commonality in explaining unsuccessful interventions or, at least, perceived barriers to success across a range of settings [28]. This has transcended health care specialism to include intervention in areas such as palliative care [3], obstetrics [4], cardiovascular health promotion [28], medication safety [29], and pharmacy auditing [30]. Hitch et al's [31] recent study on staff perceptions of an early stroke discharge system highlighted similar problems regarding its perceived compatibility. Restructuring interventions alongside the staff intended to implement them is valuable in both developing intervention *plasticity* and engagement [28–32].

When debating the potential for the CMGC intervention to be restructured, staff were forthcoming on what would benefit future modifications. For instance, the four-hour follow-up appointments measuring effectiveness of the instilled eye drop were deemed too restrictive, especially in clinics where different staff worked from morning to afternoon. The four-hour follow-ups were initially based on existing research evidence of clinical effect [33,34] balanced against the practicalities of implementing within clinics open mainly during working hours. Maximum clinical effect for the study eye drop was reported as being between 8 and 12 hours following a single dose [34]. However, the hours in which such services were open meant that asking patients to return after an eight- or 12-hour window would not be possible, potentially prompting greater strain and resistance. Balancing the intended clinical outcomes of an intervention with its flexibility has been a long-term consideration for implementation acceptability and usefulness [35], with our study highlighting its continued relevance.

For the CMGC, greater plasticity was identified in that the four-hour period could be extended up to 24 hours post-drop instillation while still offering similar clinical information [34]. Likewise, with many health boards and NHS trusts operating a six-week appointment booking system, aligning the follow-up period to mirror existing system restrictions would allow the intervention to mould more easily to local contexts. As such the algorithm for the CMGC, with modifications for the length between appointments factored in and embedding greater user discretion, may be perceived more favourably across a wider range of local contexts.

Beyond this, though, there is also a broader finding in relation to how elasticity and plasticity are perceived by those working with an intervention, and how this may influence their interactions with it. Senior staff appeared to be more likely to perceive of their clinical setting as elastic, with changes to organisational systems felt achievable. However, for other clinicians the perceived elasticity of such settings was relatively diminished, leading to concerns over the plasticity of the intervention itself. This variability in perception has been noted previously across a range of different contexts [28–32] demonstrating its salience in explaining some of the complexity associated with embedding interventions across all healthcare settings.

## Study limitations

Recruitment to the qualitative staff interviews, while achieving the study targets, would ideally have been increased to discuss emergent issues with a wider pool of participants. The issues experienced with *contextual integration* may have proved more problematic if a greater range of local contexts were introduced. That said, the variability highlighted within the sampled four clinics did provide insight into many of the broad issues that may be uncovered within any given local context. In turn, this enabled iterations of the CMGC that may prove more workable in other glaucoma clinics.

Regarding the use of NPT as a framework to understand the implementation of the CMGC, it was clear that most core constructs and subcomponents were applicable to the study data. For example, the relevance of certain subcomponents for the CMGC, such as *contextual integration*, are clearly demonstrable in the outlined findings and have contributed to a remodelling of the intervention. However, many of the subcomponents, particularly *skill set workability* and *legitimation* appear to overlap leading to potential confusion in data analysis, despite them emerging at different points in the implementation timeline [17]. Additionally, issues relating to staffing of the CMGC and its uptake, while loosely pointing towards themes of leadership and resource constraints, may have been better explained by alternative implementation frameworks where more overt solutions are offered. These could include the use of integrated knowledge translation approaches where more staff are likely to have been closely involved prior to initiation of the intervention [36], as well as the use of techniques such as change champions [37].

## Conclusions

While the CMGC intervention, as initially conceived, was viewed as overly prescriptive in some health boards, the overall value of it for staff in determining patient treatment plans and understanding their responsiveness and adherence to eye drops cannot be understated. With iterations integrated into the intervention, future research into glaucoma treatment responsiveness and adherence may benefit by integrating the remodelled CMGC into other clinics across the United Kingdom. Beyond this specialty, the study highlights the importance of local contexts for implementations and the requirement for specific organisational issues to be adequately planned for ahead of attempting to impose an intervention. We highlight particular

issues of specialist staffing and administrative processes such as appointment management which could be more easily mitigated if staff are enrolled at an earlier stage. NPT is useful for explaining these issues. Our findings demonstrated the importance of *contextual integration* and, more specifically, the plasticity of an intervention, as well as the elasticity of its intended setting.

## Acknowledgments

The research team would like to acknowledge the input of research nurse, Kathleen Price, as well as the administrative assistance of Marie Platt and Veronica Dunning.

## Author Contributions

**Conceptualization:** Simon Read, James Morgan, David Gillespie, Claire Nollett, Marjorie Weiss, Davina Allen, Pippa Anderson, Heather Waterman.

**Data curation:** Simon Read, Heather Waterman.

**Formal analysis:** Simon Read, James Morgan, David Gillespie, Claire Nollett, Marjorie Weiss, Davina Allen, Pippa Anderson, Heather Waterman.

**Funding acquisition:** James Morgan, David Gillespie, Claire Nollett, Marjorie Weiss, Davina Allen, Pippa Anderson, Heather Waterman.

**Investigation:** Simon Read, Pippa Anderson.

**Methodology:** Simon Read, David Gillespie, Claire Nollett, Davina Allen, Pippa Anderson, Heather Waterman.

**Project administration:** Simon Read, Heather Waterman.

**Validation:** Simon Read, James Morgan, David Gillespie, Claire Nollett, Marjorie Weiss, Davina Allen, Pippa Anderson, Heather Waterman.

**Writing – original draft:** Simon Read.

**Writing – review & editing:** Simon Read, James Morgan, David Gillespie, Claire Nollett, Marjorie Weiss, Davina Allen, Pippa Anderson, Heather Waterman.

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
