## [Decision Letter · Decision Letter 0]

11 May 2021

PONE-D-20-26810

Normalisation process theory and the implementation of a new glaucoma clinical pathway in hospital eye services: Perspectives of doctors, nurses and optometrists

PLOS ONE

Dear Dr. Read,

Thank you for submitting your manuscript to PLOS ONE. After careful consideration, we feel that it has merit but does not fully meet PLOS ONE’s publication criteria as it currently stands. Therefore, we invite you to submit a revised version of the manuscript that addresses the points raised during the review process.

We look forward to receiving your revised manuscript.

Kind regards,

Khatijah Lim Abdullah, DClinP, MSc., BSc

Academic Editor

PLOS ONE

Journal Requirements:

3) Please include captions for your Supporting Information files at the end of your manuscript, and update any in-text citations to match accordingly. Please see our Supporting Information guidelines for more information: http://journals.plos.org/plosone/s/supporting-information.

4)  We note that you have indicated that data from this study are available upon request. PLOS only allows data to be available upon request if there are legal or ethical restrictions on sharing data publicly. For information on unacceptable data access restrictions, please see http://journals.plos.org/plosone/s/data-availability#loc-unacceptable-data-access-restrictions.

Reviewers' comments:

Reviewer's Responses to Questions

**Comments to the Author**

1. Is the manuscript technically sound, and do the data support the conclusions?

Reviewer #1: Yes

Reviewer #2: Yes

2. Has the statistical analysis been performed appropriately and rigorously? 

Reviewer #1: N/A

Reviewer #2: Yes

3. Have the authors made all data underlying the findings in their manuscript fully available?

Reviewer #1: Yes

Reviewer #2: Yes

4. Is the manuscript presented in an intelligible fashion and written in standard English?

Reviewer #1: Yes

Reviewer #2: Yes

5. Review Comments to the Author

Reviewer #1: A well written paper applying NPT to understanding implementation feasibility for a 4hour-4week glaucoma treatment program. Some minor comments as follows:

Line 179- please clarify what does "independent of the health board" mean.

Line 191- Under clinical observations. Because it is not clear how many researchers were collecting the data it would be helpful to explain what is meant by "52 week-by-week.." means. Is it 52 consecutive weeks? Or 52 field notes? Please state if these field notes are referencing the "wider observational data" in Line 187, are they the same thing?

Line 209 "favourable ethical opinion" sounds odd, is this equivalent to ethical approval?

Line 233 "where data should sit" It is not clear what this phrase means.

Line 259 Staffing - what is the value of about mapping not having a staff to NPT constructs (in this case cognitive participation>initiation, enrolment), rather than just using a more straightforward implementation framework that would mention manpower/ staffing/ champions? This for me seems counter intuitive especially for someone more familiar with other implementation frameworks. Suggest to consider this as the NPT fit is discussed under Discussion>limitations.

Line 315- It is hard to tell who these staff are in terms of having more formal knowledge of glaucoma, perhaps some indicator of why they are considered so should be included after their quotes e.g. OH 301 (<state etc job level of position training>)

Line 397 - This theme is quite relevant as the 4 hour + 4 week schedule clearly illustrates how elasticity and plasticity can be applied to analysing feasibility. However, I wonder if the CMGC is really just about these larger parameters which were of course quite easy to predict in terms of how they would affect feasibility. It would be interesting if authors could also illustrate more macro level examples of these concepts to show how smaller instructions in the CMGC were either elastic/ plastic or not.

Line 420- " that the retention rate of patients for all procedures ran at 94.3%" Should clarify if this refers to CMGC and if so across the same period of observation as this study (ie was it a sister/ substudy)?</state>

Reviewer #2: The manuscript met all the guidelines from the introduction, methods, results, discussion and conclusion.

The study used was a non-randomised observational type with an ethnographic approch to the qualitative elements which gave a good picture of the data. The limitations were well dcumented as to the small pool of particiapnts.

6. PLOS authors have the option to publish the peer review history of their article (what does this mean?). If published, this will include your full peer review and any attached files.

Reviewer #1: **Yes: **Lee Yew Kong

Reviewer #2: **Yes: **Laly Joseph

---

## [Author Response · Author response to Decision Letter 0]

26 May 2021

Our cover letter provides more detail on our responses to the wider feedback and criteria from the journal, alongside the review feedback responses outlined below.

Reviewer 1 Line 179- please clarify what does "independent of the health board" mean. 

This phrase is indicating that the researchers conducting the fieldwork were employed by a separate institution, e.g. Cardiff University, and were independent from the day-to-day activities of the health board in which they were performing the research. We have changed the phrase to be ‘independent from the clinical team’ and believe this should enhance the clarity for readers.

Reviewer 1 Line 191- Under clinical observations. Because it is not clear how many researchers were collecting the data it would be helpful to explain what is meant by "52 week-by-week." means. Is it 52 consecutive weeks? Or 52 field notes? Please state if these field notes are referencing the "wider observational data" in Line 187, are they the same thing? 

We agree that this would benefit from further clarity. As we have established the nature of the field notes in question in the previous two sentences, we have adjusted the wording here to be: “A total of 52 of these observational field notes were taken over the duration of the study covering the majority of research visits”

Reviewer 1 Line 209 "favourable ethical opinion" sounds odd, is this equivalent to ethical approval? 

Yes – it is exactly the same but is based on the terminology of the UK-based ethical committees. Given the international readership and audience of PLOS ONE, we have amended this to ‘ethical approval’ for the purposes of this article.

Reviewer 1 Line 233 "where data should sit" It is not clear what this phrase means. 

We have adapted this sentence to further enhance clarity. This now reads ‘where data fitted within the developing themes’.

Reviewer 1 Line 259 Staffing - what is the value of about mapping not having a staff to NPT constructs (in this case cognitive participation>initiation, enrolment), rather than just using a more straightforward implementation framework that would mention manpower/ staffing/ champions? This for me seems counter intuitive especially for someone more familiar with other implementation frameworks. Suggest to consider this as the NPT fit is discussed under Discussion>limitations. 

We have added some wordings in the Study Limitations section as suggested, including reference to frameworks where implementation issues related to staffing are more clearly established, such as through the processes of knowledge translation approaches and the use of change champions during implementation etc.

Reviewer 1 Line 315- It is hard to tell who these staff are in terms of having more formal knowledge of glaucoma, perhaps some indicator of why they are considered so should be included after their quotes e.g. OH 301 () 

We have provided one or two examples within the text to highlight what is intended by this phrase. There is a risk that by aligning these to specific quotes that we enable identification of participants so believe this to be a preferable solution.

Reviewer 1 Line 397 - This theme is quite relevant as the 4 hour + 4 week schedule clearly illustrates how elasticity and plasticity can be applied to analysing feasibility. However, I wonder if the CMGC is really just about these larger parameters which were of course quite easy to predict in terms of how they would affect feasibility. It would be interesting if authors could also illustrate more macro level examples of these concepts to show how smaller instructions in the CMGC were either elastic/ plastic or not. 

We feel that we have offered a thorough discussion of this topic and highlighted several macro level examples within the Discussion section (lines 448-495 and 513-532). Given the other changes we have made above, particularly in relation to ref: 009 in the limitations section, we believe this comment is adequately covered.

Reviewer 1 Line 420- " that the retention rate of patients for all procedures ran at 94.3%" Should clarify if this refers to CMGC and if so across the same period of observation as this study (ie was it a sister/ substudy)? 

We have clarified this in-text by confirming that the retention rate was for all CMGC study procedures.

---

## [Decision Letter · Decision Letter 1]

21 Jul 2021

Normalisation process theory and the implementation of a new glaucoma clinical pathway in hospital eye services: Perspectives of doctors, nurses and optometrists

PONE-D-20-26810R1

Dear Dr. Read,

We’re pleased to inform you that your manuscript has been judged scientifically suitable for publication and will be formally accepted for publication once it meets all outstanding technical requirements.

Kind regards,

Khatijah Lim Abdullah, DClinP, MSc., BSc

Academic Editor

PLOS ONE

Reviewers' comments:

**Comments to the Author**

 All comments have been addressed

---

## [Editor Report · Acceptance letter]

23 Jul 2021

PONE-D-20-26810R1 

Normalisation process theory and the implementation of a new glaucoma clinical pathway in hospital eye services: Perspectives of doctors, nurses and optometrists 

Dear Dr. Read:

I'm pleased to inform you that your manuscript has been deemed suitable for publication in PLOS ONE. Congratulations! Your manuscript is now with our production department. 

Kind regards, 

on behalf of

Dr. Khatijah Lim Abdullah 

Academic Editor

PLOS ONE